# The Fungal Effector Mlp37347 Alters Plasmodesmata Fluxes and Enhances Susceptibility to Pathogen

**DOI:** 10.3390/microorganisms9061232

**Published:** 2021-06-06

**Authors:** Md. Saifur Rahman, Mst Hur Madina, Mélodie B. Plourde, Karen Cristine Gonçalves dos Santos, Xiaoqiang Huang, Yang Zhang, Jean-François Laliberté, Hugo Germain

**Affiliations:** 1Department of Chemistry, Biochemistry and Physics, Université du Québec à Trois-Rivières, Trois-Rivières, QC G8Z 4M3, Canada; md.saifur.rahman@uqtr.ca (M.S.R.); hmadina.geb@gmail.com (M.H.M.); melodie.bplourde@uqtr.ca (M.B.P.); cris.kgs@gmail.com (K.C.G.d.S.); 2Department of Computational Medicine and Bioinformatics, University of Michigan, 100 Washtenaw Avenue, Ann Arbor, MI 48109, USA; xiaoqiah@umich.edu (X.H.); zhng@umich.edu (Y.Z.); 3Institut National de la Recherche Scientifique-Centre Armand-Frappier Santé Biotechnologie, Laval, QC H7V 1B7, Canada; jean-francois.laliberte@iaf.inrs.ca

**Keywords:** *Melampsora larici-populina*, effector, Mlp37347, plasmodesmata, glutamate decarboxylase

## Abstract

*Melampsora larici-populina (Mlp)* is a devastating pathogen of poplar trees, causing the defoliating poplar leaf rust disease. Genomic studies have revealed that *Mlp* possesses a repertoire of 1184 small secreted proteins (SSPs), some of them being characterized as candidate effectors. However, how they promote virulence is still unclear. This study investigates the candidate effector Mlp37347’s role during infection. We developed a stable *Arabidopsis* transgenic line expressing Mlp37347 tagged with the green fluorescent protein (GFP). We found that the effector accumulated exclusively at plasmodesmata (PD). Moreover, the presence of the effector at plasmodesmata favors enhanced plasmodesmatal flux and reduced callose deposition. Transcriptome profiling and a gene ontology (GO) analysis of transgenic *Arabidopsis* plants expressing the effector revealed that the genes involved in glucan catabolic processes are up-regulated. This effector has previously been shown to interact with glutamate decarboxylase 1 (GAD1), and in silico docking analysis supported the strong binding between Mlp37347 and GAD1 in this study. In infection assays, the effector promoted *Hyalonoperospora arabidopsidis* growth but not bacterial growth. Our investigation suggests that the effector Mlp37347 targets PD in host cells and promotes parasitic growth.

## 1. Introduction

To better understand the molecular mechanisms of disease development in plants, phytopathology research has investigated the intracellular interactions between pathogens and their hosts. These studies have revealed that plants possess a large arsenal of innate immune receptors that are capable of recognizing all pathogen classes [1,2]. In order to penetrate the plant, colonize its various tissues, and cause disease, pathogens must be able to deactivate the defense responses of the plants. An essential component of pathogenesis is the secretion of small pathogenic proteins, called effectors, which modulate plant immunity and facilitate infection [3].

*Melampsora larici-poplina* (*Mlp*) is an obligate biotrophic fungal pathogen that causes poplar leaf rust. Like other rust pathogens, *Mlp* invades the plant leaves using a specialized infection structure, the haustorium, which is known to be the secretory site of effector proteins [4,5]. A combination of transcriptomic and genomic studies revealed that *Mlp* possesses an estimate of 1184 small secreted proteins (SSP) [6], also named candidate-secreted effector proteins (CSEPs). These CSEPs are expressed in the planta and inside the haustoria, but not in the spores. No specific domain or function was recognized or predicted in these CSEPs, and in most cases their sequences were specific to the order *Pucciniales* [7,8,9]. Petre et al. (2015) studied the localization and the protein interactions of 24 *Mlp* CSEPs in leaf tissue, and found them in various cellular compartments such as the nucleus, chloroplast, mitochondria, cytosolic bodies, and nucleolus. In that paper, the localization of Mlp37347 was described as cytosolic bodies, which were further demonstrated to be plasmodesmata [10]. Glutamate decarboxylase 1 (GAD1), an enzyme that catalyzes the conversion of glutamate to gamma-aminobutyric acid (GABA), was identified as an interaction partner of Mlp37347 [11].

Plasmodesmata are channels that connect the cytoplasm of neighboring plant cells, allowing the passage of small molecules such as ions, sugars, essential nutrients, and RNAs [12]. Decades of research has demonstrated that plasmodesmata dynamically regulate cell-to-cell connectivity during developmental transitions and in response to abiotic and biotic stresses. Callose, a plant β-1,3-glucan polysaccharide, is synthesized by glucan synthase-like (GSL) enzymes, also called callose synthases (CalS), and is degraded by β-1,3-glucanases (BG). It is very abundant at plasmodesmata and is often used to visualize these structures. A high level of callose deposition in the cell wall near the neck area of the plasmodesmata narrows the plasmodesma channel, whereas callose degradation opens it [13].

The investigation of the molecular basis of rust fungi pathogenicity is hampered by the fact that they cannot be cultured in vitro, nor genetically modified. Moreover, in the case of poplar leaf rust, the genetic transformation of the poplar host has so far only been possible with cultivars that are not susceptible to rust, thus preventing us to genetically manipulate both the pathogen and its host. To circumvent this hurdle and others, many groups have used heterologous systems that allow the study of effectors in planta [10,11,14,15,16,17,18,19]. In this study, we used two model plants, *Arabidopsis thaliana* and *Nicotiana benthamiana*, to study Mlp37347’s in planta activity. Our exploration of *Melampsora* candidate effector Mlp37347 revealed that it alters plasmodesmatal flux, likely through the upregulation of glucan catabolism. It also alters plant susceptibility to a filamentous pathogen, but not to a bacterial pathogen. Finally, we show that in silico protein–protein docking and in vivo yeast two-hybrid experiments strongly support the Mlp37347–GAD1 interaction.

## 2. Materials and Methods

### 2.1. Cloning Procedures and Plasmid Constructs

To obtain the nuclear localization signal (NLS)-tagged construct, the Mlp37347 ORF was amplified with a sense PCR primer encoding an NLS. The NLS-Mlp37347 amplicon was inserted by recombination in the entry vector pDONR221 and then into the expression vector pB7FWG2 using Gateway technology [20]. To generate 2×mCherry, tandem copies of mCherry were obtained as double-stranded DNA fragments (gBlocks) from Integrated DNA Technologies Inc. (Coralville, IA, USA). The 2×mCherry fragment was amplified by PCR and cloned into the pDONR221 entry vector, followed by recombination into the pK7WG2 vector using the Gateway protocol. Sequencing of all the constructs was performed before transformation in *Agrobacterium tumefaciens*.

### 2.2. Plant Material, Growth Conditions, and Transgenic Production

Seeds of *A. thaliana* and *N. benthamiana* were vernalized for 48 h at 4 °C and plants were grown in soil at 23 °C, 60% relative humidity with a 16 h/8 h light/dark cycle in a growth chamber. *Arabidopsis* Col-0 plants were transformed using *A. tumefaciens* strain C58C1 to develop transgenic plants using the modified floral dip method [21]. The GFP-tagged Mlp37347 construct and transgenic line were previously reported [10]. To select the single insertion homozygous transgenic plants, Mendelian segregation of the Basta resistance (15 mg/mL) was followed in T1, T2, and T3 generations. Two transgenic lines, Mlp37347-4 and Mlp37347-6, were used for all experiments to account for possible insertional effect. Mlp37347-6-GFP localization was previously reported in Germain et al. (2018). In the second line (line 4), effector localization (Mlp37347-4-GFP) displayed identical localization as in line 6 (Appendix A) and both transgenic lines displayed normal morphology (Appendix A). All Mlp37347 constructs lack their signal peptide.

The Salk T-DNA mutant line, Salk_022227 (*gad1–6*) was obtained from the Arabidopsis Biological Resource Center (ABRC) (Columbus, OH, USA). The lack of transcript expression has already been reported in the *gad1* mutant [22].

### 2.3. Expression in Nicotiana Benthamiana

Transient agrobacterium-mediated transformation of *N. benthamiana* leaves was performed according to the protocol of [23]. Peptone yeast extract (YEP) broth was used to grow recombinant bacterial strains overnight with appropriate antibiotics, then harvested and resuspended in infiltration buffer (10 mM MgCl_2_ and 150 μM acetosyringone). The bacterial suspensions were generally infiltrated at an OD600 of 0.5, except for the 1×mCherry and 2×mCherry constructs, which were used at an OD600 of 0.1.

### 2.4. Pathogen Infection Assays

Infection assays were done according to the methods previously described [24]. Briefly, *Pseudomonas syringae pv. tomato (Pst)* DC3000 was cultured overnight at 28 °C and infiltrated on the abaxial side of the leaves at an OD600 of 0.001. *Hyaloperonospora arabidopsidis (Hpa*) Noco2 infections were performed on 3-week-old *Arabidopsis* plants using the spray inoculation method at 20,000 spores/mL.

### 2.5. Confocal Microscopy Assay

Live-cell imaging was performed with a Leica TCS SP8 confocal laser scanning microscope (Leica Microsystems) with a 40 ×/1.40 oil immersion objective. A young leave from *A. thaliana* or *N. benthamiana* was excised into small pieces, mounted in water, and immediately observed. The GFP was excited at 488 nm, and emission was recorded at 505–530 nm. Excitation of mCherry was carried out at 552 nm, and emission was captured between 597 and 627 nm. Images were taken at 512 × 512 resolution using line-by-line scanning and using sequential scanning (when appropriate). The LAS AF Lite software (version 3.3), ImageJ, Adobe Photoshop CS6 and Illustrator were used for the post-acquisition image processing and figure assembly.

### 2.6. mCherry Diffusion Assay

The mCherry diffusion assay was performed in the leaves of *N. benthamiana* in a time-dependent manner [25]. Briefly, suspensions at an OD600 of 0.1 of Agrobacterium cells containing a plasmid for the effector and either 1× or 2×mCherry were infiltrated into the abaxial side of the leaves. Thirty-six hours later, the samples were examined by confocal laser scanning microscopy and microscope fields where single cells expressing mCherry were identified and re-imaged 4 h later to allow diffusions. The number of surrounding cells now positive for the mCherry was counted; these experiments were repeated three times. Statistical significance was calculated using the Student’s *t*-test.

### 2.7. DANS Assay and Callose Quantification 

Drop-ANd-See (DANS) dye loading assay was performed on fourth and fifth intact rosette leaves of 3-week-old *Arabidopsis* plants, as previously described by Cui [26]. Briefly, the leaves were sprayed with H_2_O_2_ and incubated at room temperature for 2 h. Subsequently, a 1 µL droplet of 1 mM carboxyfluorescein diacetate (CFDA) was loaded onto the center of the upper surface of an intact leaf, followed by confocal imaging of the abaxial surface of the washed leaf 5 min after loading CFDA dye. Confocal imaging was performed under a 40×/1.40 objective lens, using laser excitation of 488 nm with an emission of 505 to 525 nm. 

Imaging of callose deposition at plasmodesmata using aniline blue was carried out according to the protocol described by Zavaliev [27]. Confocal microscope observation was performed at 405 nm for excitation and 475 to 525 nm detection at 40×. The images were analyzed by ImageJ software using the Analyze Particle tool plug-in to quantify the amount of callose.

### 2.8. Y2H Reporter Assays

The coding sequences of Mlp37347 (without its signal peptide) and AtGAD1 or PtGAD1 were cloned into pGBKT7 (binding domain) and pGADT7 (activation domain), respectively, by homologous recombination in the yeast strain Y2H gold. Both plasmids were extracted and co-transformed in strain Y2H gold. A series of dilutions (10^−0^, 10^−1^, 10^−2^) were prepared for each transformant, and 10 µL was plated on double drop-out medium (DDO) (without Leu and Trp) and on the selective quadruple drop-out (QDO) (without Trp, Leu, His, and Ade) (Sigma-Aldrich Co., St. Louis, MO, USA) and incubated at 30 °C for 3 to 4 days for photography. 

### 2.9. Western Blot Analysis

Western blotting was carried out as described by Germain et al. [28] with minor modifications. The 3-week-old transgenic plant leaves were collected for protein extraction, and a protein aliquot was prepared. The blot was probed with an α-GFP-HRP antibody (1:500 dilution, Molecular Probes, Santa Cruz Biotechnology, Dallas, TX, USA).

### 2.10. RNA Extraction and Transcriptome Analysis

The isolation of RNA was carried out as previously described [24]. Total RNA was extracted from 2-week-old soil grown *Arabidopsis* plants and quantified before being sent for sequencing at the Genome Quebec Center of McGill University. The ontological genetic enrichment (GO) of up- and down-regulated genes (having a Q value ≤ 0.05 and a fold change ≥ 2) was studied using Cytoscape software (version 3.1.1) with the ClueGO and CluePedia plug-ins [29]. The transcriptomic data set is available in Genbank as GEO accession number GSE158410. Differential expression analysis was performed in R 3.6 with DESeq2 [30].

### 2.11. In Silico Protein–Protein Binding

The three-dimensional structures of Mlp37347 were constructed using I-TASSER [31]. The GAD1 structure (PDB ID: 3HBX) was obtained from the Protein Data Bank database (https://www.rcsb.org/ (accessed date: 6 January 2019)). The binding efficiency of Mlp37347 to GAD1 was determined using the following four different protein–protein docking host servers: ClusPro, Grammx, Patchdock, and ZDock [32,33,34,35].

## 3. Results

### 3.1. Mlp37347 Enhances Plasmodesmata Flux

The candidate effector Mlp37347 is a putative homolog of the avirulence protein AvrL567 of *Melampsora* lini [36]. In *M. larici-populina,* Mlp37347 does not have any close relatives and does not belong to a specific effector family. Besides, Mlp37347 displays a unique plasmodesmatal localization when stably expressed in *Arabidopsis* [10] and was shown to interact with *Nicotiana benthamiana* GAD1 by co-immunoprecipitation followed by mass spectrometry [11]. For the above-mentioned reasons, Mlp37347 (128 amino acids, molecular weight 15–17 kDa) was investigated as a promising candidate effector and was selected for functional studies.

To assess if Mlp37347 could alter the plasmodesmatal flux, we performed an intercellular flux assay [37] by measuring the diffusion of mCherry in the presence or absence of Mlp37347. In this assay, single and tandem mCherry (thereafter 1×mCherry and 2×mCherry, respectively) are used [38]. Because of its smaller size, 1×mCherry can diffuse into neighboring cells; however, 2×mCherry can only diffuse if the plasmodesmata is enlarged. Thirty-six hours after co-agroinfiltration, microscope fields showing a mCherry positive cell were identified. Four hours later, the same microscope fields were re-imaged, and the positive cells neighboring the initial positive cells were counted. As expected, the 1×mCherry could diffuse into neighboring cells (Figure 1A), on the other hand, the 2×mCherry did not diffuse into neighboring cells when Mlp37347 was not present, but did so when it was co-expressed with Mlp37347 (Figure 1B). These observations suggest that Mlp37347 increases the SEL of the plasmodesmata, and thus increases diffusion from cell to cell. 

### 3.2. Plasmodesmata Localization of Mlp37347 Is Required for Enhanced Plasmodesmatal Flux 

To assess if the impact of Mlp37347 on diffusion is also observed in *Arabidopsis*, we used a second complementary method, which is not limited to assessing the diffusion of a single protein, to evaluate cell-to-cell diffusion. We used a Drop-ANd-See (DANS) diffusion assay [39]. In this experiment, we also wanted to assess the importance of Mlp37347’s plasmodesmata localization in the increased diffusion. As observed in Figure 2A, the NLS-tagged protein is now segregated in the nucleus and no longer accumulated at the plasmodesmata. This DANS assay showed a significant flux increase, revealed by the larger area stained by the CF dye, in the presence of Mlp37347-6-GFP and Mlp37347-4-GFP (Figure 2B); however, this increase is not observed when the effector is restricted to the nucleus. This result demonstrates that the localization of Mlp37347 at the plasmodesmata is indeed required to increase the intercellular flux. 

### 3.3. Mlp37347–GAD1 Interaction

To assess if Mlp37347 could potentially interact with *Arabidopsis* GAD1 (its only known interactor is NbGAD1 [11]), or with GAD1 from its natural host (poplar), we used the yeast two-hybrid (Y2H) method. *Populus trichocarpa* encodes a total of five GAD, named GAD1 to GAD5. PtGAD1 is the most similar to AtGAD1, the other GAD (2, 3, 4 and 5) display 87%, 83%, 81% and 77% of amino acid identity with PtGAD1, respectively. Yeast expressing Mlp37347 in conjunction with both proteins grew on the QDO selective media (Figure 3), confirming their ability to establish a physical interaction.

To further characterize this interaction, the structure models of Mlp37347 and *Populus trichocarpa* GAD1 (PtGAD1) were constructed by I-TASSER. The predicted models of Mlp37347 and PtGAD1 have an estimated template modeling score (TM-score) of 0.85 and 0.99, respectively (Figure 4A,B), indicating the correct structural folds are confidently predicted, giving us assurance to perform the molecular docking. The experimental structure of AtGAD1 was obtained from (PDB ID: 3HBX, chain B) after removing water molecules and other ligands. The docking between Mlp37347 and AtGAD1 or PtGAD1 was performed using four different methods (ClusPro, Gramm-X, PatchDock, and Z-DOCK) [32,33,34,35]. The top-ranked poses obtained from the four approaches showed similar binding modes of Mlp37347 bound to AtGAD1 or PtGAD1 (Figure 4C; a representative pose of Mlp37347-PtGAD1 from Z-DOCK). Extensive hydrogen bonding interactions were observed between Glu46, Asn79, Arg80, and Ser104 of Mlp37347, and Arg139, Phe302, His303-Asn305, and Arg29 of PtGAD1 (Figure 4D). These hydrogen bond-forming residues in PtGAD1 are also present in AtGAD1. The sequence alignment of NbGAD1, AtGAD1, and PtGAD1 shows a high level of conservation (Appendix A), supporting a similar interaction between the effector and the three GAD1.

### 3.4. Mlp37347 Decreases Plasmodesmatal Callose Deposition and Affects Callose Metabolism Gene Expression

As the main regulatory mechanism of the plasmodesmatal flux is callose deposition, we measured callose deposition in different *A. thaliana* lines using the ImageJ software by quantifying the fluorescence of aniline blue staining. The callose was significantly reduced in both stable Mlp37347-GFP lines (Figure 5A,B) compared to Col-0. By contrast, in the transgenic expressing NLS-Mlp37347-GFP, the amount of callose deposition did not vary significantly. This result confirms our previous observation that Mlp37347’s localization to the plasmodesmata is important for its action on plasmodesmatal opening. To further probe the biological significance of the interaction between Mlp37347 and GAD1, we also measured callose deposition in a stable *gad1* knock-out line and in a cross between this *gad1* knock-out line and Mlp37347-6-GFP. In the *gad1* knock-out line, the amount of callose deposition did not vary significantly from the Col-0 control. However, the *gad1* × Mlp37347-GFP line also had a level of callose similar to the *gad1* and Col-0 lines, indicating that the reduced callose accumulation observed in the Mlp37347-GFP line is dependent on GAD1. We also tested whether the effector localization was affected by the absence of GAD1 in the *gad1* × Mlp37347-GFP line. As can be seen in Appendix A, in the absence of GAD1, the localization of Mlp37347-GFP remains unchanged. Thereby, we can conclude that the effect of the effector on callose deposition is dependent of GAD1; however, GAD1 is not required for the effector to accumulate at the plasmodesmata.

To investigate the mechanism by which Mlp37347 influences callose deposition, we performed the transcriptomic profiling of two-week--old *A. thaliana* stable transgenic seedlings expressing Mlp37347-GFP and a control plant expressing only GFP. To determine the biological processes affected by Mlp37347-GFP expression, a gene ontology (GO) terms enrichment analysis was carried out on the deregulated genes (filtered by a Q-value ≤ 0.05 and a fold change ≥ 2) using the Cytoscape software (version 3.1.1). In the Mlp37347-GFP transgenic line, 84 and 395 genes were up- and down-regulated by two-fold or greater, respectively. This analysis revealed that many genes for the catabolism of glucan (ISA3, DPE1, PHS1, PHS2) are significantly up-regulated, while some genes linked to the synthesis of glucan (GSL04, XTH19) are down-regulated in plants expressing Mlp37347-GFP (Figure 5C), reinforcing the link between Mlp37347 and plasmodesmata regulation by the control of callose deposition.

### 3.5. Mlp37347 Increases the Susceptibility of A. thaliana to H. arabidopsidis

To assess the impact of increased plasmodesmatal permeability caused by Mlp37347’s in planta expression on plant susceptibility to pathogens, we performed growth assays with two different classes of organisms: an oomycete and a bacterium. We measured the productive infection in control plants (Col-0 and Col-0-GFP) as well as in the following transgenic lines: Mlp37347-6-GFP, Mlp37347-4-GFP, NLS-Mlp37347-GFP, *gad1*, and Mlp37347-GFP × *gad1*. We observed that both lines expressing Mlp37347-GFP harbor a significantly higher susceptibility to *H. a. Noco2* (Figure 6A), whereas this increase in pathogen growth was not observed when Mlp37347 was sequestered in the nucleus (NLS-Mlp37347-GFP), nor when Mlp37347-GFP was expressed in the *gad1* line (Figure 6A). GAD1 does not seem to be directly implicated in immunity against oomycetes, as its absence does not affect *H. a. Noco2* growth.

## 4. Discussion

Several groups have recently reported that some pathogens, mainly viruses and bacteria, have proteins that associate with plasmodesmata to influence intercellular communication [40]. However, to our knowledge, a protein from rust fungi interfering with the plasmodesmatal flux has not yet been reported. Mlp37347, one of the *Melampsora larici-populina* CSEPs studied by Petre et al. in 2015 [11], and which role as a true effector has been supported by its effect on plant susceptibility to an oomycete pathogen [10], seems to be such a protein. In this study, we found that its expression in planta in the heterologous systems *A. thaliana* and *N. benthamiana* increased the plasmodesmatal diffusion rate and the plasmodesmata size exclusion limit. These observations are a probable consequence of the reduction of callose deposition in plants expressing Mlp37347. These biological effects are dependent on the plasmodesmatal localization of Mlp37347 as well as on its interaction with GAD1, both factors also being required for the increase in *H. arabidopsidis* Noco 2 infectivity observed in Mlp37347 transgenic plants. Moreover, the expression of Mlp37347 also upregulates the expression of several genes involved in glucan catabolism and downregulates some genes involved in glucan metabolism, thereby possibly explaining how Mlp37347 would exert its activity. Since Mlp37347 does not localize to the nucleus, its effects on transcription would likely be indirect.

For pathogens, an increase in plasmodesmatal flow can be favorable, as once they gain access to one cell, they could more easily draw the soluble nutrients from neighboring cells. Moreover, it has been shown that cell-to-cell propagation through plasmodesmata is used by viruses [41,42]. It is also an interesting feature for biotrophs as it could allow effectors, which are small proteins, to move through the plant to favor infection, for example by neutralizing the plant’s systemic responses. Our data indicate that the presence of Mlp37347 in a plant cell modifies the physical properties of plasmodesmata, allowing the usually size constrained 2×mCherry to move between cells (Figure 1). Moreover, the induction of beta-1,3-glucanases is also observed in a transgenic line expressing Mlp37347, as it is observed in viruses that hijack the plasmodesmata for their propagation [41]. The plasmodesmata opening caused by Mlp37347 is, however, a double-edged sword, as intercellular signaling is required for the coordination of plant defense. Plasmodesmata exploitation by fungi for intercellular propagation has been reported for *Magnaporthe oryzae* (Ascomycota), the hemibiotrophic fungus that causes the rice blast disease [43]. In their work, Kankanala reported that intracellular invasive hyphae can propagate from a cell to its neighboring cells, most likely through plasmodesmata [43]. Using a different strategy, the work of Yamaoka demonstrated that the Ascomycota *Blumeria graminis* intercellular propagation was reduced when plasmodesmata were mechanically destroyed [44]. To our knowledge, rusts have never been shown to use or manipulate plasmodesmata. Our data indicate that *Mlp* seems to manipulate plasmodesmata flux through the deregulation of callose metabolism, which is more akin to what has been observed for viruses. Our work did not, however, try to show any evidence of propagation through the plasmodesmata. 

The callose-dependent constriction of the plasmodesmata is the main regulatory mechanism of plasmodesmatal permeability [45], and is used by plants both during development and cell differentiation as well as part of the pathogen-related defense response [46,47]. The two opposite pathways, callose synthesis and callose degradation, are targeted by viruses, either directly or through transcriptional regulation [48,49]. It has already been shown that fungal effectors can also alter transcription [24,48,49]. Our transcriptional analysis of *Arabidopsis* expressing Mlp37347-GFP indicates that the genes for the catabolic processing of glucan are up-regulated, whereas the genes linked to the glucan synthesis are down-regulated. Consistently, we have observed that the amount of callose was significantly reduced in the stable Mlp37347-expressing line compared to Col-0 (Figure 5). Mlp37347’s disruption of callose deposition is, however, not sufficient to recreate the aberrant guard cells localization and proliferative clusters observed in the epithelium of plants deficient in the callose synthase GSL8 [50]. Our study establishes that the plasmodesmata localization of Mlp37347 is required for enhanced plasmodesmatal flux, as this increase was not observed when the effector was directed to the nucleus by a nuclear localization signal (Figure 2). The exact mechanism by which Mlp37347 regulates callose remains to be elucidated. Plasmodesmata functions are also regulated by the composition of their membranes, as nanodomains lipids and protein constituents are crucial for controlling the flexibility of the PD membrane, and by the cell cytoskeleton [51]. Indeed, it has been shown that some viruses increase the size exclusion limit of plasmodesmata through the depolymerization of actin filaments (F-actin) [52]. It would be interesting to see if Mlp37347 also has an impact on these plasmodesmata regulatory components. 

The two main hurdles affecting the virulence studies of poplar rust fungi are (1) rust fungi cannot be genetically manipulated, and (2) the poplar genotype that is amenable to genetic transformation is not susceptible to *Mlp*. As genetic tools are not fully available in the poplar-rust pathosystem, we investigated the virulence activity of the effector in heterologous systems. *M. larici-populina* and *Hyaloperonospora arabidopsidis* are both obligate biotrophic filamentous pathogens of dicotyledonous plants and they share similar modes of propagation through leaf tissue. Consequently, the pathogenicity of *H. arabidopsidis* would be more likely to be affected by rust effectors than the pathogenicity of the bacteria *P. syringae*. Indeed, the Mlp37347-GFP lines promoted the growth of *H. arabidopsidis*, but not to an equivalent level to the infection control line enhanced diseased susceptibility 1 (eds1), which is hyper-susceptible to *H. arabidopsidis*. Furthermore, this increase in sensitivity was not observed when Mlp37347 was sequestered in the nucleus (NLS-Mlp37347-GFP) or the *gad1* background. Meanwhile, *Pseudomonas syringae pv. tomato* DC3000 (*Pst*) bacterial infection assays showed no significant alteration in pathogen growth between genotypes, which slightly differs with the 2018 paper where the Mlp34347 plant supported slightly less bacterial growth than the controls. Xian recently showed that the RipI effector of *Ralstonia solanacearum* promotes the interaction of GADs with calmodulin, increasing the production of GABA. They also demonstrated that *R. solanacearum* can replicate efficiently using GABA as a nutrient, and that RipI and plant GABA contribute to a successful infection [53]. These observations from Xian might in part explain why Mlp37347 is unable to promote infection in *gad1* plants: rather than promoting GAD1 interaction with calmodulin Mlp37347, it could negatively affect it.

Based on our results, we drew a hypothetical model (Figure 7). After being secreted from *M. larici-populina*, the effector Mlp37347 would distribute in the cell and accumulate at the plasmodesmata, and to a lesser extent in the cytosol. Since GAD1 has been reported to be a cytosolic protein, the interaction between GAD1 and Mlp37347 could occur either in the cytosol or at the plasmodesmata. Indirectly, the cellular presence of Mlp37347 causes a decrease in the expression of genes involved in glucan metabolism and concurrently an increased expression of genes involved in glucan catabolism, resulting in the opening of the plasmodesmata (depicted in Figure 7). To uncover the precise mechanism by which Mlp37347 promotes plant susceptibility will require further experiments.

This study expands our molecular knowledge of the poplar *Melampsora larici-populina* pathosystem and highlights the importance of effector target proteins in plant susceptibility. Moreover, it provides the first line of evidence that rust fungal pathogens can exploit the plasmodesmata. Future work will be aimed at unraveling the specific mechanisms by which Mlp37347 affects the plant defense.

## Figures and Tables

**Figure 1 microorganisms-09-01232-f001:**
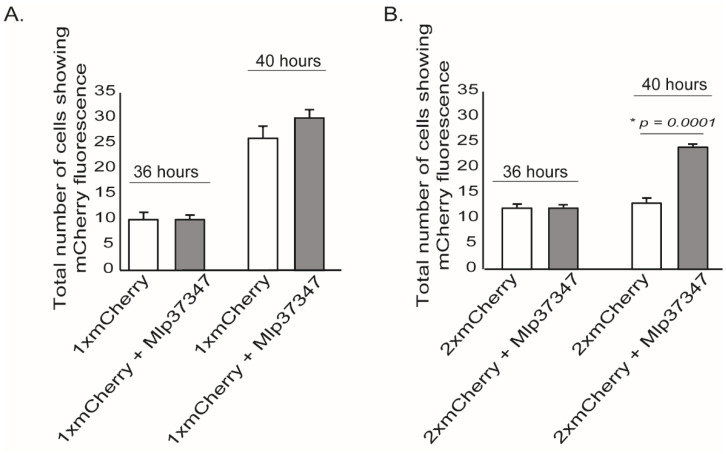
Mlp37347 enhances plasmodesmata flux. (**A**) Quantification of 1×mCherry diffusion to surrounding cells with or without the effector provided a measure of molecular flux through plasmodesmata. (**B**) Quantification of 2×mCherry diffusion. Data represent mean ± SD. The Student’s *t*-test determined the statistical difference from the control leaves without Mlp37347-GFP; asterisks indicate statistical significance. Experiments were repeated three times (*N* = 3).

**Figure 2 microorganisms-09-01232-f002:**
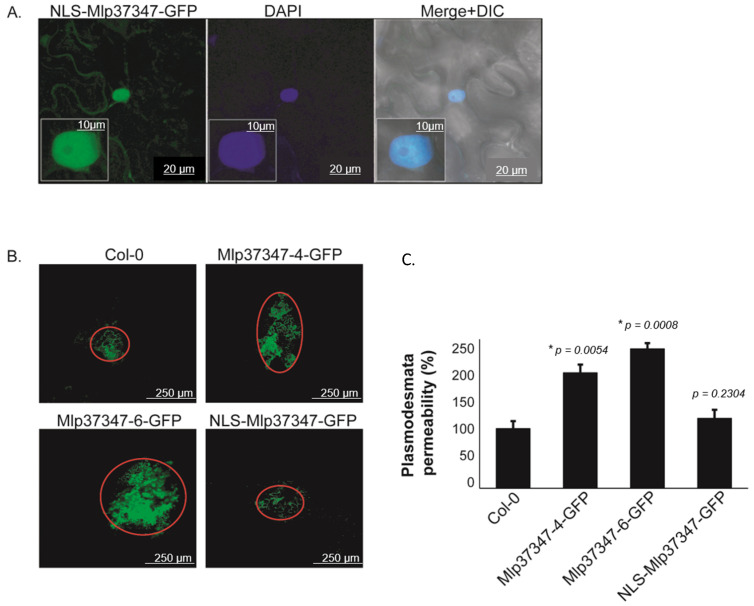
Plasmodesmata localization of Mlp37347 is required for enhanced plasmodesmatal flux. (**A**) Subcellular accumulation of NLS-Mlp37347-GFP in *N. benthamiana* epidermal cells at 3 days post-infiltration, the nucleus was stained by DAPI, and epidermal cells were observed under the green channel (**left panel**), blue channel (**middle panel**), and merge of the two channels +DIC (**right panel**). (**B**) DANS experiments were performed on the fifth and sixth leaves of three-week-old *A. thaliana* plants and imaged with the confocal microscope. (**C**) Permeabilities were measured in percentage compared to control. Data represent mean ± SD. The statistical difference from the control Col-0 leaves was determined by Student’s *t*-test, * *p*-values indicate statistical significance. Experiments were repeated at least three times (*N* = 3), and representative data are shown.

**Figure 3 microorganisms-09-01232-f003:**
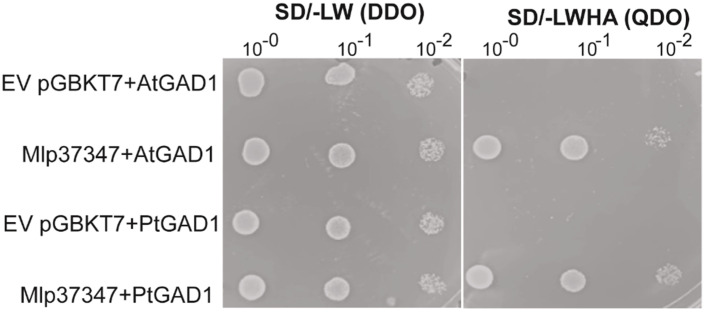
Mlp37347 interacts with AtGAD1 and PtGAD1. Co-expression of AtGAD1/PtGAD1 with Mlp37347 in yeast shows interaction between GAD1 and Mlp37347. Yeast co-expressing the indicated combination of bait and prey were spotted on the synthetic double dropout medium lacking leucine and tryptophan (SD/-LW (DDO)) and quadruple dropout medium lacking leucine, tryptophan, histidine, and adenine (SD/-LWHA (QDO)). The plates were photographed 3–4 days after inoculation.

**Figure 4 microorganisms-09-01232-f004:**
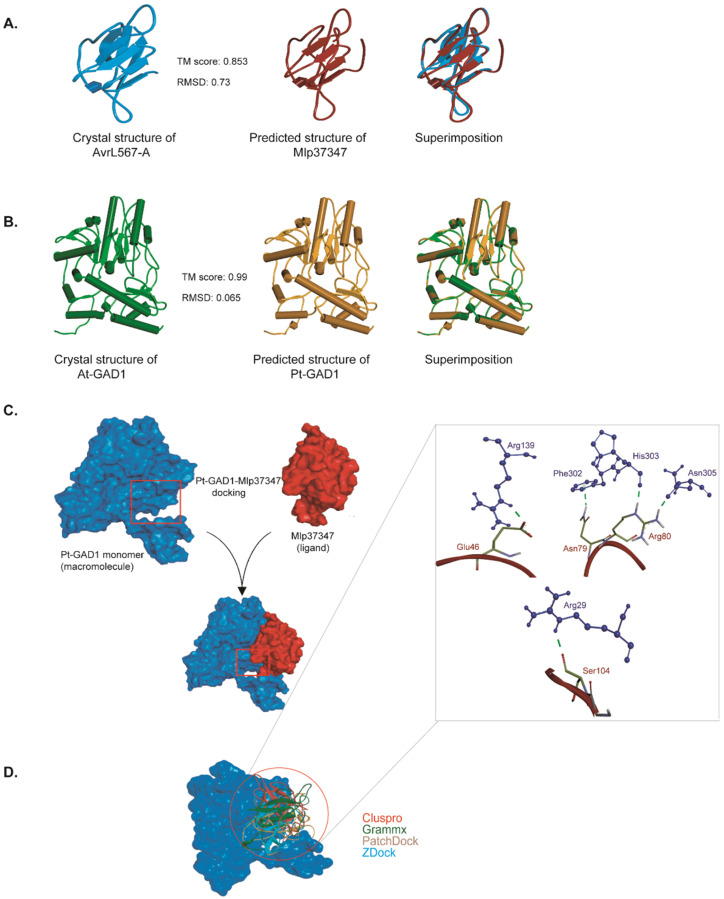
Molecular modeling of Mlp37347–GAD1 interaction. (**A**,**B**) Predicted structures of Mlp37347 and Populus trichocarpa GAD1. TM-score ranges from zero to one, where one indicates a perfect match between two structures. (**C**) The general scheme of docking between AtGAD1 (blue) and Mlp37347 (red) using different servers. (**D**) Assembly of top results from different servers. On the right panel, a close-up view of the hydrogen bonding network orientation of the PtGAD1–Mlp37347 complex. GAD1 and Mlp37347 are shown as blue and red, respectively.

**Figure 5 microorganisms-09-01232-f005:**
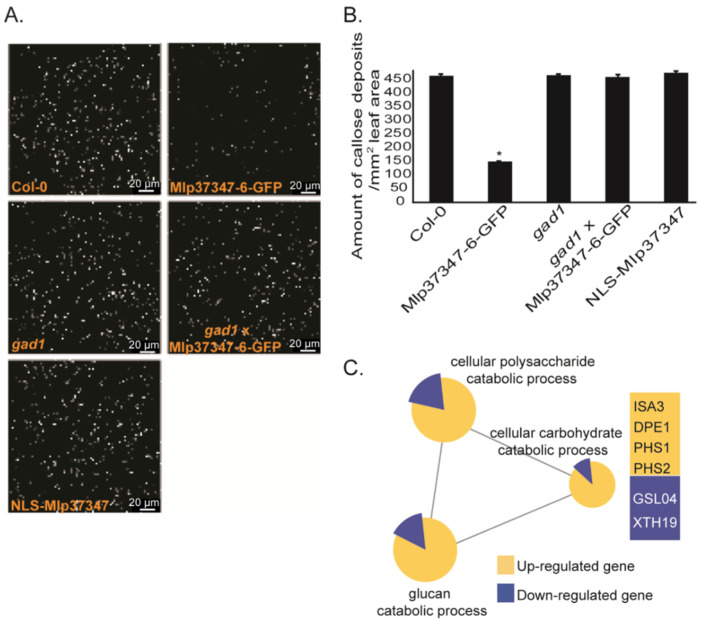
Mlp37347 decreases plasmodesmata callose deposition and affects callose metabolism gene expression. (**A**) Aniline blue staining to visualize the callose levels in *A. thaliana* transgenic lines Mlp37347-GFP, NLS-Mlp37347-GFP, *gad1*, *gad1* × Mlp37347-GFP and in Col-0 as a control. Images of callose deposition were taken at 40×/1.40 magnification, excitation at 488 nm with an emission of 505 to 550 nm using a Leica SP8 confocal microscope. (**B**) Quantification of callose deposition in the different lines. Data represent mean ± SD. The statistical difference from the control Col-0 leaves was determined by Student’s *t*-test, asterisks indicate statistical significance. (**C**) Transcriptional changes induced by the expression of Mlp37347-6-GFP. Term enrichment analysis was performed on deregulated genes (Q-value ≤ 0.05, fold change ≥ 2) using the Cytoscape software (version 3.1.1).

**Figure 6 microorganisms-09-01232-f006:**
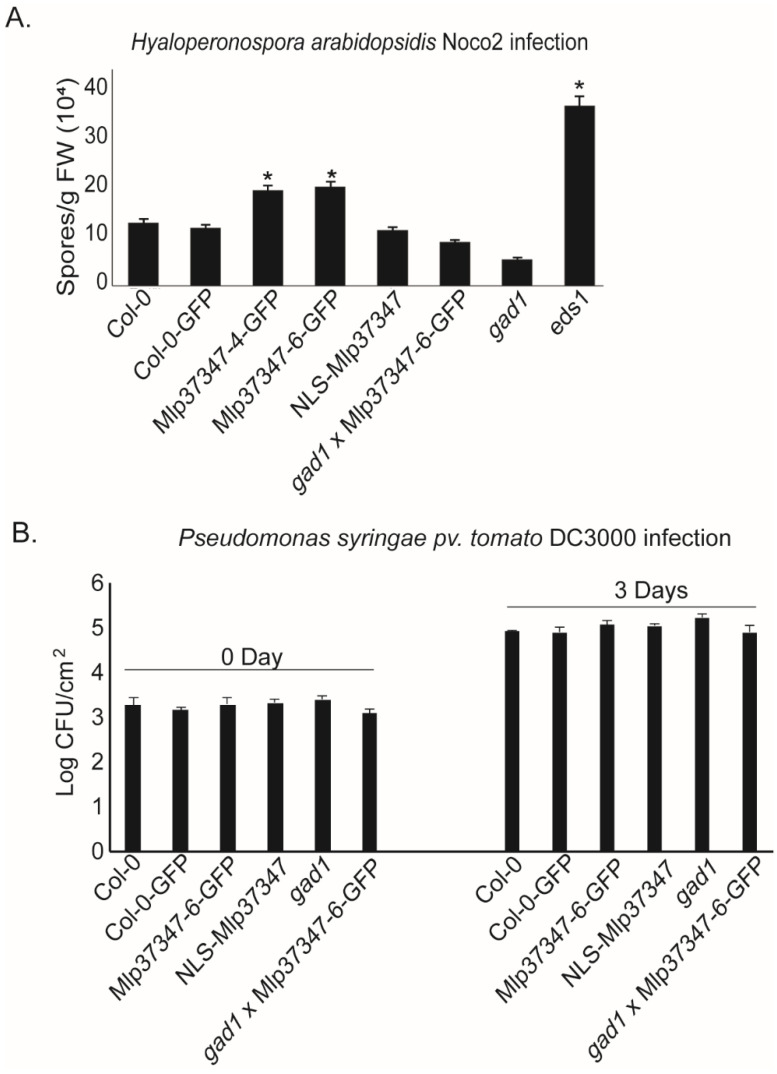
Mlp37347 increases the susceptibility of *A. thaliana* to *H. arabidopsidis*. (**A**) Growth of *Hyaloperonospora arabidopsidis* Noco2 inoculated at 20,000 conidiospores/mL. The number of spores/g fresh weight was quantified seven days after inoculation. Bars represent the mean of four replicates. Statistical significance with Col-0 was evaluated using Student’s *t*-test (*p* < 0.05) and was denoted as an asterisk. (**B**) The number of colony forming units of *Pst* DC3000 was measured on day 0 and day 3 after infection of 4-week-old soil-grown plants by leaf infiltration. A bacterial suspension with OD600 = 0.001 was used as inoculum. Statistical significance was evaluated using Student’s *t*-test (*p* < 0.05).

**Figure 7 microorganisms-09-01232-f007:**
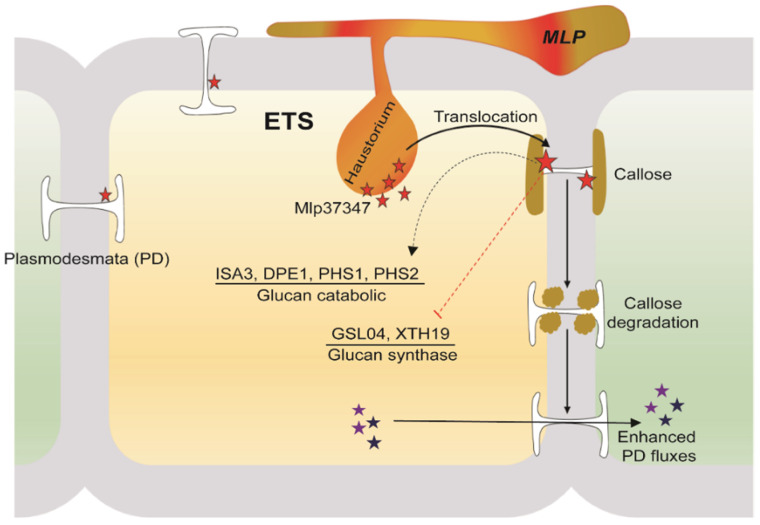
A hypothetical model illustrating a potential role of Mlp37347 on plasmodesmata during the infection.

## Data Availability

The data that support the findings of this study are openly available in GENBANK, reference number GEO accession number: GSE158410.

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
