# Peer review of "The Fungal Effector Mlp37347 Alters Plasmodesmata Fluxes and Enhances Susceptibility to Pathogen"

_microorganisms, 2021, doi:10.3390/microorganisms9061232_

Round 1
Reviewer 1 Report
Title: The fungal effector Mlp37347 alters plasmodesmata fluxes and en- 2 hances susceptibility to pathogen
Authors: Md. Saifur Rahman1 , Mst Hur Madina1 , Mélodie B. Plourde1 , Karen Cristien Gonclaves dos Santos1 , Xiaoqiang 4 Huang2 , Yang Zhang2 , Jean-François Laliberté3 and Hugo Germain*1
In this work the role of candidate effector Mlp37347 detected on Melampsora larici-populina, was investigated developping stable Arabidopsis transgenic line. The work is interesting and useful for molecular basis of rust fungi pathogenicity investigation. However some parts should be improved to increase understanding of the manoscript. Especially in the materials and methods, and results sections. Concerning the materials and methods section, the method is described in detail but the workflow should be better explained. Concerning the results section, they must be rewritten with more clarity by removing excessive methodological explanations and / or citations that will be included in the discussion. Verify the journal guideline for a correct inclusion of the references in the text. Other comments are mentioned below.
Abstract: change Mlp Mlp
Line 59: change ‘….(GSL) (also called Callose synthases (CalS)) enzymes and I’ with
‘(GSL), also called Callose synthases (CalS) enzymes
Line 79: change ‘was PCR amplified with a sense PCR primer’ with ‘ was amplified with a sense PCR primer’
Line 86: change ‘Agrobacterium tumefaciens.’ With ‘Agrobacterium tumefaciens.’
Check all Latin names in the text and change them to italics format
Line 96: ‘Germain et al. 2018’ modify the citation in the text according to the journal procedures. Then modify the following ones
Line 97: ‘Mlp37347-4-GFP for which the localization in indistinguishable to Mlp37347-6-GFP’
No clear
Lines 100-102: ‘ The Salk T-DNA mutant line, Salk_022227 (gad1-6) was obtained from Arabidopsis Bi- 100 ologicaI Resource Center (ABRC), Columbus OH, USA. The presence of aberrant tran- 101 script was already reported in the gad1 mutant [22]
It is not clear what it means here
Line 116: ‘Confocal microscopy’ modify with ‘Confocal microscopy assay’ or words that make it clear why you do it. GFP assay?
Line 159: ‘by Germain [28] ‘ add et al
Lines 190-192: ‘To assess if Mlp37347 could alter the plasmodesmatal flux, we performed an inter- 190 cellular flux assay [37] by measuring the diffusion of mCherry in the presence or absence of Mlp37347. In this assay single and a tandem mCherry (thereafter 1XmCherry and 192 2XmCherry) is used [38]. ‘
I suggest to including this sentence in the materials and methods. It is not an results
Figure 1: on the x-axis use of abbreviations of names
Lines 211.213: ‘To assess if the impact of Mlp37347 on diffusion is also observed in Arabidopsis, we 211 used a second complementary method to evaluate cell-to-cell diffusion which is not lim- 212 ited to assessing the diffusion of a single protein’
Also, this sentence for me is more appropriate in materials and methods section. It’s important for to understand the workflow. It is an aim, not a result
Lines 234-238: ‘To dissect the molecular mechanisms responsible for the effect of Mlp37347-GFP on 234 plasmodesmatal flux, we focused our attention on its known interactor, GAD1. Petre et 235 al. (2015) showed Mlp37347 interacts with N. benthamiana (NbGAD1) [11]. To assess if 236 Mlp37347 could potentially interact with Arabidopsis GAD1 or with GAD1 from its natural 237 host (poplar), we used the yeast-two hybrid (Y2H) method to test the interaction between 238 AtGAD1 or PtGAD1 and Mlp37347.’
Also here they are too many explanations not relate to yours results . In general, the flow of work should be written better, and in a more concise manner, indicating the aims on materials and methods section. Here it is difficult to understand the result.
Line 235: modify ‘Petre et al. (2015) showed Mlp37347 interacts with N. benthamiana (NbGAD1) [11].’ With ‘Petre et al. [11] showed Mlp37347 interacts with N. benthamiana (NbGAD1) [11]. this sentence is up for discussion
Line 251: ‘In silico approaches are of crucial importance in the evaluation of protein-protein 251 interactions [40].’
see above
Lines330-337: ‘These results suggest that both the locali- 330 zation of the effector at the plasmodesmata and its interaction with GAD1 are necessary 331 for Mlp37347 to increase the susceptibility of Arabidopsis to the oomycete Hyaloperenospora 332 arabidopsidis. In contrast, Pseudomonas syringae pv. tomato DC3000 bacterial infection assays 333 showed no alteration in the pathogen growth in any of the genotypes tested (Figure 6B). 334 From these experiments, we conclude that Mlp37347 promotes the growth of the filamen- 335 tous fungi-like pathogen H. arabidopsidis, but not of the bacterial pathogen P. syringae in 336 A. thaliana’
This part is for discussion or for conclusion section.
Figure 6: on the x-axis use of abbreviations of names
Line 352: ‘by Petre in 2015, [11]’
change according to the journal guideline
Line 237: ‘ cells (Figure 1). it is not necessary to cite the figure in the discussion
Line 397: ‘ cells (Figure 5). it is not necessary to cite the figure in the discussion
Author Response
In this work the role of candidate effector Mlp37347 detected on Melampsora larici-populina, was investigated developping stable Arabidopsis transgenic line. The work is interesting and useful for molecular basis of rust fungi pathogenicity investigation. However some parts should be improved to increase understanding of the manoscript. Especially in the materials and methods, and results sections. Concerning the materials and methods section, the method is described in detail but the workflow should be better explained. Concerning the results section, they must be rewritten with more clarity by removing excessive methodological explanations and / or citations that will be included in the discussion. Verify the journal guideline for a correct inclusion of the references in the text. Other comments are mentioned below.
Q1. Abstract: change Mlp Mlp
R1. Done
Q2. Line 59: change ‘….(GSL) (also called Callose synthases (CalS)) enzymes and I’ with
‘(GSL), also called Callose synthases (CalS) enzymes
R2. Changed as suggested.
Q3. Line 79: change ‘was PCR amplified with a sense PCR primer’ with ‘ was amplified with a sense PCR primer’
R3. Changed as suggested.
Q4. Line 86: change ‘Agrobacterium tumefaciens.’ With ‘Agrobacterium tumefaciens.’
R4. Changed as suggested.
Q5. Check all Latin names in the text and change them to italics format
R5. Other instances were found and changed.
Q6. Line 96: ‘Germain et al. 2018’ modify the citation in the text according to the journal procedures. Then modify the following ones
R6. Done
Q7. Line 97: ‘Mlp37347-4-GFP for which the localization in indistinguishable to Mlp37347-6-GFP’
R7. We agree that this sentence was not clear, we replaced by the following. Two transgenic lines, Mlp37347-4 and Mlp37347-6 were used for all experiments to account for possible insertional effect. Mlp37347-6-GFP localization was previously reported in Germain et al. (2018). In the second line (line 4), effector localization (Mlp37347-4-GFP) displayed identical localization as in line 6 (Supplementary figure 1A) and both transgenic lines displayed normal morphology (Supplementary figure 1B).
Q8. Lines 100-102: ‘ The Salk T-DNA mutant line, Salk_022227 (gad1-6) was obtained from Arabidopsis Bi- 100 ologicaI Resource Center (ABRC), Columbus OH, USA. The presence of aberrant tran- 101 script was already reported in the gad1 mutant [22]
It is not clear what it means here
R8. We modified to the following sentence: The lack of transcript expression was already reported in the gad1 mutant [22].
Q9. Line 116: ‘Confocal microscopy’ modify with ‘Confocal microscopy assay’ or words that make it clear why you do it. GFP assay?
R9. Changed as suggested
Q10. Line 159: ‘by Germain [28] ‘ add et al
R10. Done as suggested.
Q11. Lines 190-192: ‘To assess if Mlp37347 could alter the plasmodesmatal flux, we performed an inter- 190 cellular flux assay [37] by measuring the diffusion of mCherry in the presence or absence of Mlp37347. In this assay single and a tandem mCherry (thereafter 1XmCherry and 192 2XmCherry) is used [38]. ‘
I suggest to including this sentence in the materials and methods. It is not an results
R11. We agree it is not a result but it introduces the result and explains to the reader why we used this specific approach. Moving this sentence to the Mat and Met would confuse the reader since it wouldn’t be clear for which result each method was used. We disagree to the change, however, if the editor suggest otherwise, or if this is a journal style issue, we are open to review our position.
Q12. Figure 1: on the x-axis use of abbreviations of names
R12. We believe using full names is clearer. We prefer not to change it, unless the editor instructs otherwise.
Q13. Lines 211.213: ‘To assess if the impact of Mlp37347 on diffusion is also observed in Arabidopsis, we 211 used a second complementary method to evaluate cell-to-cell diffusion which is not lim- 212 ited to assessing the diffusion of a single protein’
Also, this sentence for me is more appropriate in materials and methods section. It’s important for to understand the workflow. It is an aim, not a result
R13. Similarly to what we mentioned to Query 11, we believe that is it important to explain why we used a different method. We did not make the suggested change.
Q14. Lines 234-238: ‘To dissect the molecular mechanisms responsible for the effect of Mlp37347-GFP on 234 plasmodesmatal flux, we focused our attention on its known interactor, GAD1. Petre et 235 al. (2015) showed Mlp37347 interacts with N. benthamiana (NbGAD1) [11]. To assess if 236 Mlp37347 could potentially interact with Arabidopsis GAD1 or with GAD1 from its natural 237 host (poplar), we used the yeast-two hybrid (Y2H) method to test the interaction between 238 AtGAD1 or PtGAD1 and Mlp37347.’
Also here they are too many explanations not relate to yours results . In general, the flow of work should be written better, and in a more concise manner, indicating the aims on materials and methods section. Here it is difficult to understand the result.
R14. Here we agree that the intro of the method was too long, we reduced by 75%. The section removed is show below.
To dissect the molecular mechanisms responsible for the effect of Mlp37347-GFP on plasmodesmatal flux, we focused our attention on its known interactor, GAD1. Petre et al. (2015) showed Mlp37347 interacts with N. benthamiana (NbGAD1) [11]. To assess if Mlp37347 could potentially interact with Arabidopsis GAD1 (its only known interactor (11) or with GAD1 from its natural host (poplar), we used the yeast-two hybrid (Y2H) method to test the interaction between AtGAD1 or PtGAD1 and Mlp37347
Q15. Line 235: modify ‘Petre et al. (2015) showed Mlp37347 interacts with N. benthamiana (NbGAD1) [11].’ With ‘Petre et al. [11] showed Mlp37347 interacts with N. benthamiana (NbGAD1) [11]. this sentence is up for discussion
R15. Since we remove the section this Query no longer applies.
Q16. Line 251: ‘In silico approaches are of crucial importance in the evaluation of protein-protein 251 interactions [40].
R16. We agree to remove this sentence.
Q17. Lines330-337: ‘These results suggest that both the locali- 330 zation of the effector at the plasmodesmata and its interaction with GAD1 are necessary 331 for Mlp37347 to increase the susceptibility of Arabidopsis to the oomycete Hyaloperenospora 332 arabidopsidis. In contrast, Pseudomonas syringae pv. tomato DC3000 bacterial infection assays 333 showed no alteration in the pathogen growth in any of the genotypes tested (Figure 6B). 334 From these experiments, we conclude that Mlp37347 promotes the growth of the filamen- 335 tous fungi-like pathogen H. arabidopsidis, but not of the bacterial pathogen P. syringae in 336 A. thaliana’
This part is for discussion or for conclusion section.
R17. We agree with the reviewer. This section was removed
Q18. Figure 6: on the x-axis use of abbreviations of names
R18. We believe using full names is clearer. We prefer not to change it, unless the editor instructs otherwise.
Q19. Line 352: ‘by Petre in 2015, [11]’
R19. et al was added
Q20. Line 237: ‘ cells (Figure 1). it is not necessary to cite the figure in the discussion
R20. Although citing the figure is not mandatory we believe it improves clarity. We did not modify the text.
Q21. Line 397: ‘ cells (Figure 5). it is not necessary to cite the figure in the discussion
R21. Although citing the figure is not mandatory we believe it improves clarity. We did not modify the text.
Reviewer 2 Report
In the manuscript “The fungal effector Mlp37347 alters plasmodesmata fluxes and enhances susceptibility to pathogen“, the authors analyze, in planta, function of the virulence-promoting small protein Mlp37347, secreted by the rust fungus Melampsora larici-populina, which is a problematic poplar pathogen. Regarding difficulties with Melamspora and poplar genetic manipulation, the authors applied heterogenic expression of Mlp37347 in Arabidopsis thaliana and found that the fungal protein enhance the plasmodesmatal flux and reduce callose deposition. Moreover, Mlp37347 interacts with the A. thaliana glutamate decarboxylase 1 (GAD1) and promotes expression of genes involved in the glucan catabolism. These profound changes seems to be responsible for the systemic colonization of plant host by not only Mlp, but also by other filamentous non-fungal pathogens.
In my opinion, the topic analyzed here is of interest for the readers of Microorganisms and it is important to understand generally how filamentous pathogens break plant resistance in systemic infections. I have a few suggestions, which should improve the clarity of the manuscript, as listed below:
Due to section order in this journal, Materials and Methods are at the beginning, and it would be better to explain in each paragraph what for was used these methods. It would clarify the reading and comprehension.
Reading previous work of the authors it is know that Mlp37347 is GFP-tagged at the end of protein and Mlp37347 is shortened, without secretion leader sequence, albeit it would be useful to clarify this also in this manuscript. Moreover, in supplementary files, the list of plasmids, their maps and primers would be helpful for comprehension.
In Discussion it would be good to speculate the function of GAD and how is involved in expression of GH. Maybe its product, GABA regulates GH-related genes expression? (PMID: 19704616)
Author Response
In the manuscript “The fungal effector Mlp37347 alters plasmodesmata fluxes and enhances susceptibility to pathogen“, the authors analyze, in planta, function of the virulence-promoting small protein Mlp37347, secreted by the rust fungus Melampsora larici-populina, which is a problematic poplar pathogen. Regarding difficulties with Melamspora and poplar genetic manipulation, the authors applied heterogenic expression of Mlp37347 in Arabidopsis thaliana and found that the fungal protein enhance the plasmodesmatal flux and reduce callose deposition. Moreover, Mlp37347 interacts with the A. thaliana glutamate decarboxylase 1 (GAD1) and promotes expression of genes involved in the glucan catabolism. These profound changes seems to be responsible for the systemic colonization of plant host by not only Mlp, but also by other filamentous non-fungal pathogens.
In my opinion, the topic analyzed here is of interest for the readers of Microorganisms and it is important to understand generally how filamentous pathogens break plant resistance in systemic infections. I have a few suggestions, which should improve the clarity of the manuscript, as listed below:
Q1. Due to section order in this journal, Materials and Methods are at the beginning, and it would be better to explain in each paragraph what for was used these methods. It would clarify the reading and comprehension.
R1. Since the other reviewer actually suggested the opposite, that we remove the little explanation of why we chose each method, we try to get a balance approached, we removed some explanation that Rev1 suggested and kept and made clearer other explanations.
Q2. Reading previous work of the authors it is know that Mlp37347 is GFP-tagged at the end of protein and Mlp37347 is shortened, without secretion leader sequence, albeit it would be useful to clarify this also in this manuscript. Moreover, in supplementary files, the list of plasmids, their maps and primers would be helpful for comprehension.
R2. Very good point. We now mentioned this in the Material and methods.
Q3. In Discussion it would be good to speculate the function of GAD and how is involved in expression of GH. Maybe its product, GABA regulates GH-related genes expression? (PMID: 19704616)
R3. Here we believe that we do not have sufficient data to speculate or formulate hypothesis on the role of GABA in the process analysed in our manuscript.